# MASK R-CNN FOR AUTOMATED MULTI-SPECIES MALARIA PARASITE DETECTION

## ABSTRACT

This study investigates the automatic detection and segmentation of malaria parasites across various Plasmodium species using Mask R-CNN, an advanced deep-learning architecture. Expanding on earlier studies in digital malaria diagnosis, we apply pixel-level segmentation to overcome the drawbacks of previous approaches. 971 microscopic pictures of four Plasmodium species—P. falciparum, P. malariae, P. ovale, and P. vivax—taken from Rwanda's healthcare facilities make up our dataset. This dataset was used to train the Mask R-CNN model, which produced excellent mean average precision (mAP) scores for all species, with P. vivax and P. malariae showing the most excellent performance with mAP 0.9575 and mAP 0.9459, respectively. Compared to earlier techniques, this method shows notable advances in parasite localization and delineation, suggesting the possibility of more precise and effective malaria diagnosis in clinical settings.

## 1 INTRODUCTION

Malaria, a mosquito-borne infection caused by Plasmodium parasites, poses a significant threat to global health, particularly in developing nations. This disease, transmitted through the bites of infected Anopheles mosquitoes, wreaks havoc on individual lives and communities. Four Plasmodium parasites, P.falciparum, P. malariae, P. ovale and P. vivax, are known to cause malaria, with P. falciparum and P. vivax responsible for most cases and demonstrating the highest levels of virulence. In 2021 alone, the World Health Organization (WHO) reported an estimated 241 million cases, with a staggering 95% concentrated in Africa. Furthermore, 627,000 lives were tragically claimed by this disease globally, with 96% of these deaths occurring in 29 countries (WHO, 2021). Recently, various studies have been on automating the malaria diagnosis process, focusing on the microscopy approach. Microscopic blood smears remain the gold standard for malaria diagnosis but are inconvenient. Traditional malaria diagnosis involves preparing thin or thick blood smears and visually examining them under a microscope. Thick smears help detect the presence of parasites, while thin smears help identify the parasite species and stage (Makhija et al., 2015). However, this process takes time and relies heavily on operator skills, making it impractical in areas with limited resources. Rapid diagnostic tests offer a more accessible and efficient alternative (Kigozi et al., 2021). This study builds upon existing research on digital systems for malaria monitoring (Mukamakuza et al., 2022). The more extensive study aims to streamline data collection, analysis, treatment, and public information services, potentially improving malaria forecasting, management, and treatment. In line with this goal, Mary et al. (2023) developed a data management model for Rwanda. Other research for this paper has investigated a few computer vision deep learning models for segmentation and detection of parasites in 2D photos (Akpo et al., 2024; Karasira et al., 2024; Bogale et al., 2024). This paper focuses on enhancing malaria prediction and automating the diagnosis process to improve malaria control efforts further using the Mask R-CNN.

## 2 LITERATURE REVIEW

Convolutional Neural Networks and Deep learning are advancing in the race for efficient automated malaria diagnosis. Their ability to automatically extract robust features from images leads to high performance, making it a frontrunner compared to other feature extraction techniques (Litjens et al., 2017; LeCun et al., 2015). A growing body of research, detailed below, showcases the effectiveness

of applying deep learning algorithms to identify malaria parasites from microscopic images of thin and thick blood smears.

A seminal paper by Poostchi et al. (2018) laid the groundwork for employing advanced image analysis and machine learning techniques in malaria detection. This research highlighted the pivotal role of these modern information technologies in effectively combating the disease. The proposed methods incorporated diverse techniques encompassing image processing, cell segmentation, parasite identification, and feature calculation, demonstrating the vast potential of AI-driven approaches in this domain.

A study by Krishnadas et al. (2022) employed YOLOv4 and YOLOv5 object detection models, achieving impressive accuracy rates of 83% and 78.5%, respectively. This suggests the potential for these deep learning algorithms to aid medical professionals in accurate identification and stage prediction of malaria, a critical step in managing the disease. While much of the research has focused on Plasmodium Falciparum due to its prevalence and lethality in sub-Saharan Africa, as acknowledged by the WHO (**?**), promising advancements are being made in diversifying the approaches to encompass other Plasmodium species. This broadened scope holds immense potential for improving overall malaria diagnosis and control.

In another study, the two-stage malaria parasite detection model proposed by Yang et al. (2020) demonstrates limitations. The initial stage, tasked with selecting potential parasite candidates using grayscale image intensity values, proves inadequate due to the inherent shortcomings of traditional image processing techniques. These techniques struggle to adapt to varying environmental conditions, resulting in inaccurate parasite identification. Consequently, the performance of the subsequent CNN classifier suffers, ultimately hindering the overall effectiveness of the model.

For this work, (Bogale et al., 2024) used Faster R-CNN with various backbone architectures and discovered that ResNet-50 produced the best results. This study highlighted the potential for CNN-based models to improve diagnostic speed and accuracy. Another study for this same project looked at YOLOv5 for parasite identification (Karasira et al., 2024). While this method effectively detected most parasites, it struggled with P. falciparum due to dataset restrictions. On the same project, (Akpo et al., 2024) investigated U-Net segmentation for malaria parasites and obtained impressive accuracy rates. However, the F1-score suggested that there was still space for improvement in recognizing positive cases, highlighting the crucial need for larger datasets, particularly for Plasmodium falciparum. As a result, the dataset section of this study discusses an enhanced dataset collection.

Building upon this foundation of promising research, this study explores the application of Mask R-CNN, a robust deep-learning architecture, for automated malaria parasite detection and segmentation. Mask R-CNN stands out for its ability to identify parasites and generate pixel-level masks, providing precise localization and delineation of individual parasites within the image. This capability presents significant advantages for accurate diagnosis and parasite quantification, potentially leading to improved patient management and treatment outcomes(He et al., 2017).

The rest of this paper is structured as follows: The architecture description 3,then The Detection, which includes the description of the extensive datasets acquisition and preprocessing and training covered in Section 4; the results and discussion are covered in Section 5 followed by the challenges and future section 6 and the conclusions are covered in Section 7.

## 3 MASK R-CNN

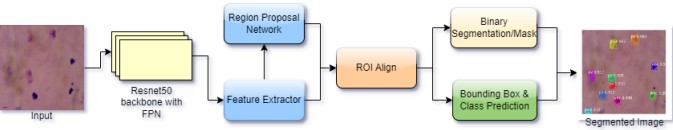

Figure 1: Mask R-CNN Architecture

Convolutional Neural Networks (CNNs), known for their prowess in computer vision, are making waves in the medical field. Their ability to analyze massive amounts of medical images opens

doors to unprecedented advancements in image analysis, particularly in cell biology experiments. This translates to increased efficiency and accuracy, paving the way for improved medical research and diagnosis (Bote-Curiel et al., 2019; Zemouri et al., 2019). One of the critical strengths of Convolutional Neural Networks (CNNs) is their modularity and flexibility. This allows existing, pre-trained networks to be readily adapted and applied to new tasks by leveraging relevant labelled images. This process, known as Transfer Learning, allows for rapid development of new applications with minimal effort. By seamlessly integrating prior knowledge with new data, transfer Learning enhances the capabilities and adaptability of CNNs, making them even more valuable in diverse domains (Yao et al., 2019).

CNN has progressed so much that researchers have used faster R-CNN to localize and classify malaria-infected cells. Faster R-CNN is a type of Region-based CNN (R-CNN). This versatile object detection model utilizes a two-stage approach. The first stage, the Region Proposal Network (RPN), identifies potential object locations within the image. Drawing inspiration from Faster R-CNN, the second stage extracts features from each proposed region using a RoIPool technique. This information is then employed for both object classification and bounding box refinement. The network utilizes shared features for both stages, leading to faster inference (Ren et al., 2015). While Faster R-CNN demonstrates impressive capabilities in object detection, it suffers from the limitation of generating coarse bounding boxes. This can lead to overlapping boxes, where multiple boxes enclose the same object, potentially hindering the accuracy and clarity of object detection. This limitation becomes particularly significant when dealing with complex images containing numerous objects or objects with intricate shapes (Hung et al., 2017).

Recognizing the limitations of coarse bounding boxes in Faster R-CNN, Mask R-CNN emerges as a powerful solution. This advanced architecture builds upon Faster R-CNN's foundation while introducing a crucial innovation: a third branch dedicated to object mask prediction. This branch operates concurrently with the branches for object detection and bounding box regression, enabling Mask R-CNN to perform instance segmentation alongside object detection (Litjens et al., 2017)

Unlike Faster R-CNN's coarse boxes, Mask R-CNN generates high-quality masks that precisely delineate the shape and boundaries of individual objects in the image. Furthermore, Mask R-CNN's architecture facilitates efficient and straightforward training, and the model demonstrates remarkable adaptability to diverse applications and tasks.

## 4 THE DETECTION METHOD

### 4.1 DATASETS, ANNOTATION AND PREPROCESSING

The dataset encompasses an extensive assortment of microscopic images that depict four distinct types of malaria parasites: Plasmodium Falciparum, Plasmodium Malariae, Plasmodium Ovale, and Plasmodium Vivax. These images were meticulously acquired at the Rwanda Biomedical Centre (RBC) Mbituyumuremyi (2024) through a specialized microscope arrangement. This setup involved giemsa-stained slides observed under a microscope, equipped with a camera attached to the eyepiece and linked to a laptop, as illustrated in Figure 2 below.

During the adjustment of the microscope, digital images (fields) of the slides were captured and stored for subsequent analysis. The images encompassed thick and thin film smears, offering a diverse array of samples for examination. Each image underwent meticulous annotation using the VGG Image Annotator 2.0.12 to ensure precision in identifying infected regions. This tool facilitated accurate delineation of infected areas through its polygon feature. An image mostly consists of parasites, white blood cells or other artefacts. In each image, the white blood cells and parasites were all labelled.

The dataset comprises images of four Plasmodium species responsible for malaria infection in humans. Specifically, it contains 278 images of Plasmodium falciparum (P. falciparum), 258 images of Plasmodium malariae (P. malariae), 260 images of Plasmodium ovale (P. ovale), and 175 images of Plasmodium vivax (P. vivax). This distribution ensures a relatively balanced representation of the major malaria-causing parasites, with a slightly lower number of P. vivax images compared to the other species. To optimize the dataset for training while balancing computational resources, all photos were preprocessed. This technique included auto-orientation and uniform scaling to size of

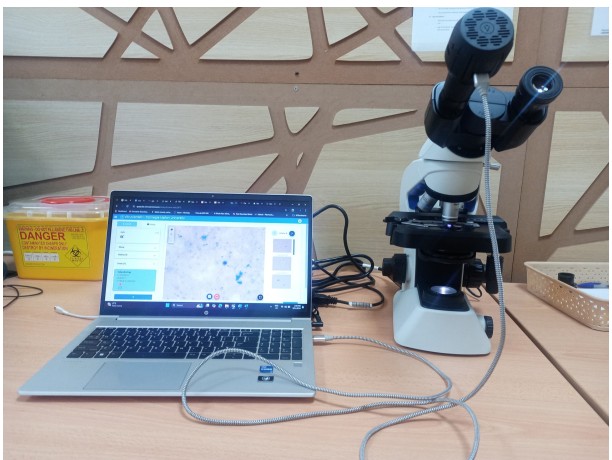

Figure 2: Camera mounted on microscope for digital data collection.

$256 \times 256$ pixels. This standardization serves as a compromise between keeping image quality and detail and managing the computing needs of the training process within the constraints of available resources and dataset size.

Strategic partitioning of the dataset was undertaken to enable practical training, validation, and testing of the proposed model on different Plasmodium infections. An 70% allocation was designated for training, 20% for validation and 10% for testing. This allocation ensures a comprehensive representation of each parasite species across distinct subsets. The dataset is a robust foundation for researching malaria parasites and developing proficient machine-learning models for their identification and classification

This study makes use of clinical samples collected from individuals who presented with a high fever at Rwandan healthcare facilities. As part of its quarterly quality control process, the Rwandan Biomedical Centre (RBC) gathers positive blood slides for additional investigation, which includes our research. This sample collection is overseen by the RBC, which has the legal ability to perform quality control, research, and training projects in Rwanda. The established system for sample collecting and utilization follows ethical norms and applicable regulations. This process occurs in the context of routine healthcare service and quality assurance, which justifies the inclusion of these samples in our analysis. Our research is integrated into existing healthcare and quality control systems, which ensures that the study benefits from real-world clinical samples while upholding ethical standards and regulatory compliance.

### 4.2 TRAINING

For this study, several experiments were conducted on four different parasites and on all of them combined to test for mixed infections. The Stochastic Gradient Descent (SGD) was utilized for each parasite dataset with a learning rate of 0.01. The training used a batch size of 8, as we classified images into three or more classes. The classes for each experiment included the background, parasites, and white blood cells. With the preprocessing described before, augmentation was not included in this experiment as it reduced the quality of the results.

The implementation was carried out using PyTorch, with the model parameters and optimizer configured accordingly. The hyperparameters included a learning rate of 0.01, momentum of 0.9, and weight decay of 0.0005. The training process was conducted over 100 epochs using a CUDA-enabled GPU. A learning rate scheduler (StepLR) was implemented with a step size of 5 and gamma of 0.1 to adjust the learning rate during training. The training loop involved iterating over the training dataset for the specified number of epochs, with periodic validation on a separate validation set.

The study used the mean average precision (mAP) statistic to assess its performance in object detection and knowledge retrieval tasks. This statistic uses a confusion matrix to compare accurate

Table 1: mAP results on Experiments

| Experiment | Validation mAP | Test mAP |
|---|---|---|
| PF | 0.7174 | **0.7737** |
| PM | 0.8547 | **0.9459** |
| PO | 0.8357 | **0.8620** |
| PV | 0.9462 | **0.9575** |
| Combined | 0.8759 | **0.8915** |

and erroneous classifications. It also uses Intersection over Union (IoU) to assess the accuracy of object localization by comparing actual and predicted bounding boxes. Furthermore, precision and recall measures evaluate the accuracy and quantity of optimistic forecasts. By combining these components, mAP accurately assesses a model's ability to locate and identify items inside images

## 5 RESULTS AND DISCUSSION

### 5.1 RESULTS

In this section, we present the results of our experiments on parasites. Table 1 shows the number of epochs, number of classes, and mAP values for each experiment.The figures that follow depict sample detection images for the different experiments and tests.

### 5.2 DISCUSSION

The results of our study demonstrate the effectiveness of Mask R-CNN in automated malaria parasite detection and segmentation. By leveraging this advanced deep learning architecture, we have addressed several limitations of previous methods and achieved promising results across different Plasmodium species.

The capacity of Mask R-CNN to precisely segment and detect malaria parasites marks a breakthrough in automated diagnosis. Mask R-CNN creates pixel-level masks that precisely define parasite borders, unlike earlier models like Faster R-CNN, which produce coarse bounding boxes. This level of precision is essential in microscopy-based diagnosis, where parasites are frequently mistaken for other blood components or abnormalities. Our findings demonstrate that all trials had high mAP values, with P. vivax (0.9575) and P. malariae (0.9459) performing very well on test data. These results imply that Mask R-CNN can handle morphological differences between Plasmodium species well, which is crucial for correct diagnosis.

The two-stage parasite detection methodology described by previous researchers Poostchi et al. (2018), suffered from environmental heterogeneity, resulting in erroneous parasite identification. Our Mask R-CNN solution overcomes this restriction by learning robust features directly from image data rather than depending on traditional image processing approaches for initial candidate selection. This end-to-end learning process enables the model to adapt to changing image situations, resulting in more accurate parasite detection across various samples.

Furthermore, Mask R-CNN's segmentation capacity outperforms approaches that depend purely on classification or bounding box detection. Our technique improves individual parasite localization and delineation in the image by providing exact masks for each detected parasite.

The training options outlined earlier were critical in balancing performance and computational restrictions, mainly using a pretrained Mask R-CNN model and image scaling to 256x256 pixels. While the small batch size of 8 and the lack of data augmentation may limit sample diversity and exposure to different picture circumstances, these decisions prioritized training stability and retaining real microscopy features. Despite these limits, our model displayed significant generalizability among Plasmodium species. This accomplishment implies that our technique efficiently used transfer learning and negotiated the obstacles given by restricted computational resources, resulting in a robust model for malaria parasite detection.

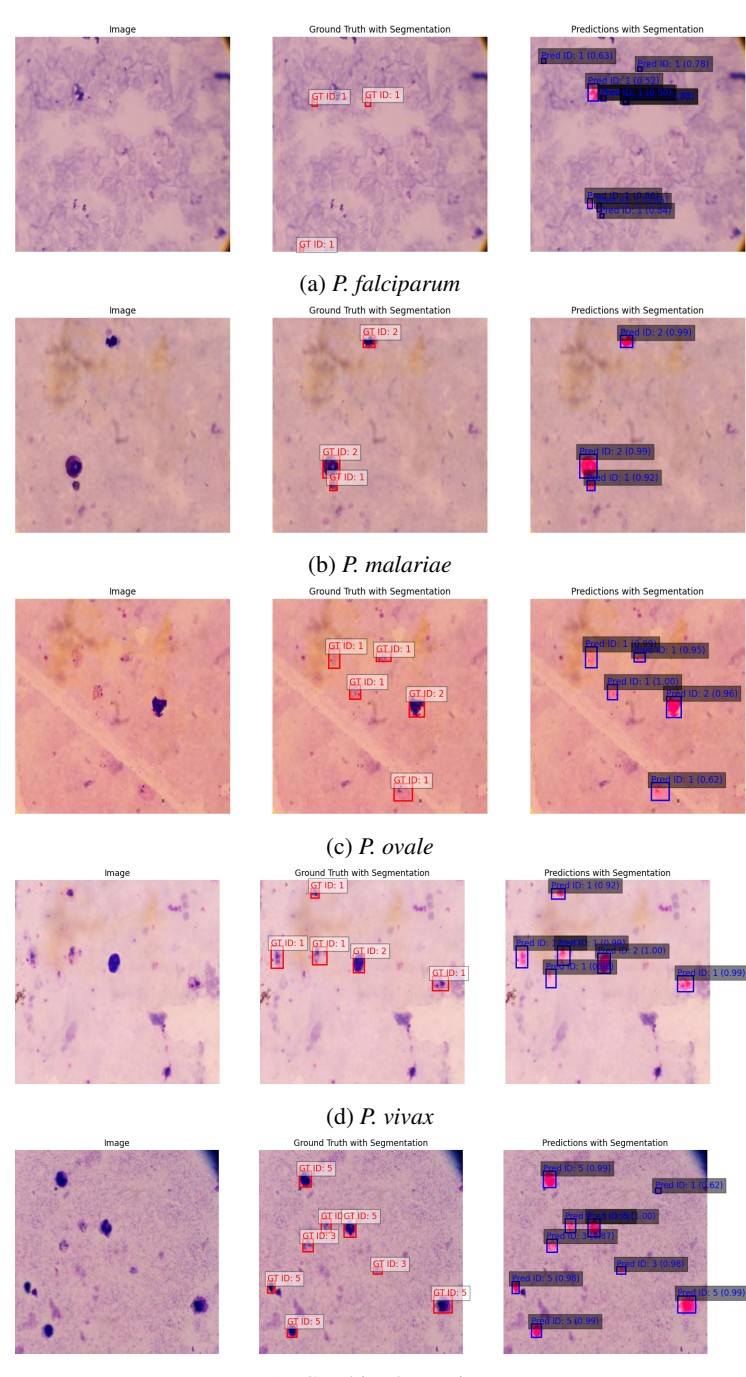

Figure 3: Sample test images of parasites detected in blood smears for different species.

Our Mask R-CNN model outperforms earlier deep-learning methods for detecting malaria parasites. Unlike YOLOv5, which struggled to recognize P. falciparum due to a small dataset, our model performs well across all species. Mask R-CNN's capacity to handle mixed infections and complicated pictures and its pixel-level segmentation enables exact parasite location. This allows automated staging and species identification in subsequent iterations, making it a powerful tool for increasing malaria diagnosis accuracy and efficiency.

## 6    FUTURE WORK AND RECOMMENDATIONS

Our work, while intriguing, has limitations that indicate topics for future investigation. To improve the model's performance and applicability, we make many recommendations.

First and foremost, extending the dataset is critical, particularly for underrepresented species and parasite stages. A larger dataset will boost model performance, especially for uncommon species or unconventional appearances. Second, increased computational resources allow for bigger batch sizes and more extensive hyperparameter adjustment, potentially improving the model's learning capacity and generalizability.

Furthermore, carefully planned data augmentation approaches can strengthen the model's resilience to fluctuations in microscope pictures. Furthermore, expanding the model to include information regarding parasite morphology and developmental stages could be a helpful addition, building on the accurate detection and segmentation accomplished in this study.

Additionally, real-world clinical validation is required to assess the model's practical applicability and identify areas for development in various healthcare settings. Creating techniques for interpreting and explaining the model's decisions helps boost trust and adoption among healthcare practitioners.

Finally, longitudinal studies can evaluate the model's long-term effectiveness and ability to adjust to parasite shape or prevalence alterations. Finally, incorporating the model into current healthcare systems can help it gain traction and significantly influence malaria detection and treatment.

By resolving these issues, we may build on the foundation established by this study and progress in the automated malaria parasite detection and diagnosis field.

## 7    CONCLUSION

Our research illustrates the effectiveness of Mask R-CNN for automated malaria parasite detection and segmentation across several Plasmodium species. The model's capacity to achieve high accuracy and precise localization even in the presence of morphological differences and mixed illnesses marks a significant step forward in microscopy-based diagnosis. Despite the promising results, this study has limitations. Future research should focus on increasing the dataset, particularly for underrepresented species and stages, and performing real-world clinical validations to evaluate the model's effectiveness in various scenarios. In conclusion, our Mask R-CNN technique is promising for boosting malaria diagnosis and control efforts. By solving the challenges of microscopy-based parasite identification, this technique has the potential to influence global healthcare systems significantly.

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
