# OpenReview forum: "Mask R-CNN for Automated Multi-Species Malaria Parasite Detection"
_ICLR.cc/2025/Conference — Submitted to ICLR 2025_

### Official Review · Reviewer_sudT · 2024-11-01

**Soundness:** 2
**Presentation:** 2
**Contribution:** 1
**Rating:** 1
**Confidence:** 5

**Summary:**

This study investigates the automatic detection and segmentation of malaria parasites across various Plasmodium species using Mask R-CNN.

**Strengths:**

The application of Mask RCNN may be new.

**Weaknesses:**

The is just like a project report, not ready for conferences like ICLR.

Mask R-CNN is 7 years old, there is no novelty of the work in terms of methodology.

The introduction is very simple, CNN and R_CNN etc are known by default and there is no need to , it seems they don't have the knowledge of recent advances of AI,  e.g. YOLOv5 is outdated.

There are no comparisons with other detection/segmentation method, and no ablation studies.

**Questions:**

What GPU used?

Please define PM, PO etc in the table. Could you also provide the IoU results?

Please check references, some are missing such as WHO (?)

**Details Of Ethics Concerns:**

Details of ethical approvals are not provided.

---

### Official Review · Reviewer_Nxib · 2024-11-04

**Soundness:** 2
**Presentation:** 3
**Contribution:** 1
**Rating:** 3
**Confidence:** 5

**Summary:**

The authors collect and annotate around a thousand microscopic pictures of malaria parasite variants. Then, they apply Mask-RCNN to this dataset to detect the parasites and obtain a good mAP score.

**Strengths:**

The dataset gathered and annotated is quite large for a medical dataset, which is excellent work and sure to be useful to those who can make use of it in the future. Although I assume it will not be possible to make the dataset open due to privacy concerns.
Furthermore, detection results appear to be good, i.e. the mAP score is decent.

**Weaknesses:**

There is too little novelty to be had. The authors are essentially applying an existing model to a new dataset and obtaining good results. Applying existing models to their own problems is what good engineers do all around the world, all the time. While the paper's results may be valuable for people in the medical domain interested in the state-of-the-art for malaria parasite detection, I do not believe it is of interest to the machine learning community at large.
Furthermore, it's even difficult to place the results as state-of-the-art, because no comparisons are made with previous models. Some are touched upon in the literature review section, but in terms of quantifiable results, we only have test and validation mAP for Mask-RCNN. How can we know that Mask-RCNN performed better than the other approaches with only this information? The verbal claims need to be substantiated with further evidence.
There is also no interesting domain-specific analysis which could give us insights into why the dataset is interesting, or why other models struggled while Mask-RCNN was good. Or even where Mask-RCNN seemed to be failing. We only find nuggets of common knowledge which suggest that the model performs "very well" due to reasons such as it being "end-to-end", and among possible improvements are gathering a larger dataset and training with a more powerful GPU.

**Questions:**

I have addressed my questions in the "Weaknesses" section above.

---

### Official Review · Reviewer_8iiT · 2024-11-04

**Soundness:** 2
**Presentation:** 2
**Contribution:** 1
**Rating:** 1
**Confidence:** 5

**Summary:**

The paper tackles the detection of malaria parasites in blood smears with mask R-CNN.

**Strengths:**

Easy to read paper, tackling a relevant societal problem.

**Weaknesses:**

The paper lacks sufficient novelty. It is a straightforward application of mask R-CNN to malaria parasite detection without any adaptation of the methodology.

Addtionally, and although the authors state "... this method shows notable advances...", there is no empirical comparison with SOTA or baseline methods, no ablation  studies.

The dataset is also not properly described, not being clear about the difficulty of the task (for instance, class distribution, distribution of the number of parasites per image, etc.).

**Questions:**

none

---

### Meta-Review · Area_Chair_zRuN · 2024-12-14

**Metareview:**

Dear authors,

Thank you for submitting the draft. The reviewers' rankings indicated that the draft is not ready for publication at this stage.

Authors claims to investigate "automatic detection and segmentation of malaria parasites across various Plasmodium species". Main concern was novelty, as reviewers noted it was "a straightforward application of mask R-CNN to malaria parasite detection". There were also concerns about claim of the state-of-the-art. Reviewers suggested that authors should try new models. Authors did not engage the reviewers.

We hope comments by reviewers will help improve the draft.

regards

AC

**Additional Comments On Reviewer Discussion:**

All reviewers agree draft is not ready for publication.

---

### Decision · Program_Chairs · 2025-01-22

Reject